# DIscBIO: A User-Friendly Pipeline for Biomarker Discovery in Single-Cell Transcriptomics

**DOI:** 10.3390/ijms22031399

**Published:** 2021-01-30

**Authors:** Salim Ghannoum, Waldir Leoncio Netto, Damiano Fantini, Benjamin Ragan-Kelley, Amirabbas Parizadeh, Emma Jonasson, Anders Ståhlberg, Hesso Farhan, Alvaro Köhn-Luque

**Affiliations:** 1Department of Molecular Medicine, Institute of Basic Medical Sciences, University of Oslo, 0372 Oslo, Norway; seyedamirabbas.parizadeh@medisin.uio.no (A.P.); hesso.farhan@medisin.uio.no (H.F.); 2Oslo Centre for Biostatistics and Epidemiology, Faculty of Medicine, University of Oslo, 0372 Oslo, Norway; w.l.netto@medisin.uio.no; 3Department of Urology, Northwestern University, Chicago, IL 60611, USA; damiano.fantini@gmail.com; 4Simula Research Laboratory, 1325 Lysaker, Norway; benjaminrk@simula.no; 5Sahlgrenska Center for Cancer Research, Department of Laboratory Medicine, Institute of Biomedicine, Sahlgrenska Academy at University of Gothenburg, SE-41390 Gothenburg, Sweden; emma.jonasson@llcr.med.gu.se (E.J.); anders.stahlberg@gu.se (A.S.); 6Wallenberg Centre for Molecular and Translational Medicine, University of Gothenburg, SE-41390 Gothenburg, Sweden; 7Department of Clinical Genetics and Genomics, Sahlgrenska University Hospital, SE-41390 Gothenburg, Sweden

**Keywords:** single-cell sequencing, normalization, gene filtering, ERCC spike-ins, biomarkers, DEGs, decision trees, network analysis, Jupyter notebook, binder

## Abstract

The growing attention toward the benefits of single-cell RNA sequencing (scRNA-seq) is leading to a myriad of computational packages for the analysis of different aspects of scRNA-seq data. For researchers without advanced programing skills, it is very challenging to combine several packages in order to perform the desired analysis in a simple and reproducible way. Here we present DIscBIO, an open-source, multi-algorithmic pipeline for easy, efficient and reproducible analysis of cellular sub-populations at the transcriptomic level. The pipeline integrates multiple scRNA-seq packages and allows biomarker discovery with decision trees and gene enrichment analysis in a network context using single-cell sequencing read counts through clustering and differential analysis. DIscBIO is freely available as an R package. It can be run either in command-line mode or through a user-friendly computational pipeline using Jupyter notebooks. We showcase all pipeline features using two scRNA-seq datasets. The first dataset consists of circulating tumor cells from patients with breast cancer. The second one is a cell cycle regulation dataset in myxoid liposarcoma. All analyses are available as notebooks that integrate in a sequential narrative R code with explanatory text and output data and images. R users can use the notebooks to understand the different steps of the pipeline and will guide them to explore their scRNA-seq data. We also provide a cloud version using Binder that allows the execution of the pipeline without the need of downloading R, Jupyter or any of the packages used by the pipeline. The cloud version can serve as a tutorial for training purposes, especially for those that are not R users or have limited programing skills. However, in order to do meaningful scRNA-seq analyses, all users will need to understand the implemented methods and their possible options and limitations.

## 1. Introduction

Single-cell RNA sequencing (scRNA-seq) is a powerful technology that has already shown great potential and the field is rapidly expanding [1,2]. In animals, scRNA-seq is providing a leap forward in resolving cellular diversity and giving unprecedented insight into gene expression changes during development, regeneration and disease [3,4,5,6,7,8]. Recently, scRNA-seq also started to flourish in plant research [9,10]. The growing attention toward single-cell transcriptomics calls for the development of computational tools to interactively analyze scRNA-seq data for any target organism. Multiple tools for scRNA-seq analysis are being developed [11,12]. Some tools, such as Seurat [13], SC3 [14], ASAP [15], Granatum [16] and SC1 [17], provide integrated pipelines for single-cell data. However, many others are characterized by specific analytic scopes in the wide spectrum of scRNA-seq analysis [18], thus leading many researchers to integrate several tools to address their questions.

RaceID is an excellent tool developed by Grün and colleagues to perform different aspects of scRNA-seq data analysis, including preprocessing, sub-population identification by k-means clustering, outlier cells detection and assessing differences in gene expression across clusters [19,20]. On the other hand, TSCAN is a remarkable tool for in silico pseudo-time ordering implemented over cell clusters resulted mainly from model-based clustering [21]. Both RaceID and TSCAN do not handle external RNA controls consortium (ERCC) spike-in, which can be used for accounting for cell-to-cell technical differences [22,23]. Furthermore, they lack features for biomarker discovery, such as networking, decision trees and gene enrichment analysis. The biomarker discovery approach enables researchers to find molecular markers and gene expression signatures for better diagnostic or prognostic techniques. Moreover, it could guide therapeutic decision-making [24]. Thus, multiple research efforts have been directed towards biomarker discovery and a number of computational tools are available. For instance, STRING is an online resource for networking analysis with known and predicted protein–protein interactions and functional enrichment analysis [25]. EnrichR is a prominent tool developed in the Ma’ayan lab for inferring knowledge about an input gene set by comparing it to annotated gene sets from over 160 libraries [26]. Decision trees have been extensively applied for the identification of biomarkers [27,28]. Decision trees are transparent and interpretable predictive models but they require considerable programming skills [29]. The RWeka package is a collection of machine learning algorithms, including decision tree analysis [30]. The rpart package is widely used for generating recursive partitioning trees [31]. For researchers with limited to no programming skills it can be very challenging to combine several of these tools to be able to perform a complete analysis. This is especially true for the designing of a complex computational pipeline that goes from single-cell sequencing read counts to biomarker discovery. Moreover, due to the required concatenation of heterogeneous programs and custom scripts via file-based inputs and outputs, as well as the program dependencies and version requirements, such a pipeline may suffer from reproducibility problems [32,33].

Hence, we developed DIscBIO (DIscovery of single-cell transcriptomics BIOmarkers). DIscBIO is an open-source, multi-algorithmic pipeline that provides an opportunity to analyze scRNA-seq data for any organism with a taxonomy ID. It allows biomarker discovery using decision trees and gene enrichment analysis in a network context from single-cell sequencing read counts through clustering and differential analysis. DIscBIO is implemented as an R package [34] published on the Comprehensive R Archive Network (CRAN). It can be run either in command-line mode or through a computational pipeline using Jupyter notebooks [35]. DIscBIO allows users to conveniently construct, analyze, visualize and tune scRNA-seq data interactively. DIscBIO notebooks integrate R code with explanatory text and output data and images in a sequential narrative. The notebooks allow R users to understand the different steps of the pipeline and learn how to apply it to analyze their own data. Importantly, they do not need to worry about the non-trivial task of connecting the different packages. However, in order to do meaningful analyses, they still need to understand the implemented methods, their possible options or parameters and their limitations. We believe that users with some programing skills will be able to tune individual parameters while more experienced programmers can fully edit and extend the pipeline to their needs. We also provide a cloud version using Binder [36] that allows users with very limited programming skills to run the pipeline even without installing the programming language R, Jupyter, or any of the software that the pipeline integrates. In order to produce the cloud-hosted Binder version, the repository includes a specification of all the software used, which enables automatic or manual reproduction or modification of the necessary computational environment needed to execute the pipeline on any computational resources [36]. DIscBIO facilitates the creation of publication-ready plots for researchers with all levels of programming proficiency. 

DIscBIO is open source, and it is freely available at https://github.com/ocbe-uio/DIscBIO. Its R package can be downloaded from https://cran.r-project.org/package=DIscBIO. To fully harness the potential of transcriptome analysis in deciphering complex diseases at the single-cell level, we showcase DIscBIO over two scRNA-seq datasets. The first dataset consists of circulating tumor cells (CTCs) from patients with breast cancer, the second one is from a myxoid liposarcoma cell line. In the first case, we investigate the connection between the Golgi apparatus and oncogenesis of breast cancer to demonstrate how our analysis pipeline can be used to study this link based on current knowledge. In the second case, we focus on defining the molecular signature of the sub-populations of myxoid liposarcoma cells in association with their cell cycle phases. Additionally, we showcase the analysis of a mouse scRNA-seq dataset with accession number GSE41265 [37] to demonstrate how to load and analyze datasets from the conquer repository [11]. The conquer repository provides access to analysis-ready scRNA-seq datasets from different species, including humans, mice and zebrafish. 

## 2. Pipeline Description

DIscBIO is a multi-algorithmic pipeline for an easy, fast and efficient analysis of sub-populations and the molecular signatures that characterize them. The pipeline consists of four successive steps: (1) data pre-processing; (2) cellular clustering and pseudo-temporal ordering; (3) determining differentially expressed genes (DEGs); and (4) biomarker identification, including decision trees, networking and gene enrichment analysis (Figure 1). Below, we summarize the sections and algorithms used in the pipeline. Detailed information may also be found in each of the sections of the notebooks:

### 2.1. Data Pre-Processing

Prior to applying data analysis, normalization and gene filtering are used to pre-process the raw read counts that resulted from the sequencing. To account for RNA composition and sequencing depth among samples (single cells), the normalization method “median of ratios” is used through RaceID. This method makes it possible to compare the normalized counts for each gene equally between samples because it takes the ratio of the gene instantaneous median to the total counts for all genes in that cell. The key idea of gene filtering is to highlight the genes that manifest high variation across samples. DIscBIO provides three gene filtering techniques: noise filtering, expression filtering and designed filtering. In case the data include ERCC spike-ins, genes can be filtered based on variability in comparison to a noise level estimated from the ERCCs using an algorithm developed by Brennecke et al. [22]. For datasets with or without ERCCs, genes can be filtered either based on their gene expression or minimum expression in a certain number of cells, or based on a particular gene list of interest, for instance the genes involved in a desired cellular process. The expression filtering is computed through RaceID using two parameters: Minexpr and Minnumber.

### 2.2. Cellular Clustering and Pseudo-Temporal Ordering

Cellular clustering is performed according to gene expression profiles to detect cellular sub-populations with unique features. DIscBIO allows k-means clustering [38], using the RaceID algorithm, and model-based clustering [39], using TSCAN software. This pipeline estimates the number of clusters by finding the minimal cluster number at the saturation level of the gap statistics, which standardizes the within-cluster dispersion [40]. DIscBIO enables a robustness assessment of the detected clusters in terms of stability and consistency using Jaccard’s similarity statistics and silhouette coefficients [41,42]. To visualize the detected clusters, two common dimensionality reduction tools are implemented: principal component analysis (PCA) and *t*-distributed stochastic neighbor embedding (*t*SNE). DIscBIO uses the “tsne” package through RaceID to plot the clusters in *t*SNE maps [43,44]. Additionally, DIscBIO can detect outlier cells. The outlier identification is implemented based on RaceID using a background model based on the distribution of the transcript counts within a cluster. The background model is computed using the mean and the variance of the expression of each gene in a cluster. Outliers are defined as cells with a minimum number of outlier genes. This number is set by default to 2. In the case studies we set it to be 5% of the number of genes in the filtered dataset; this is based on the recommendation of De Vienne et al. [45]. Outlier genes are inferred from non-normalized transcript counts. Finally, pseudo-temporal ordering is implemented over the clusters resulting from the k-means and model-based clustering using TSCAN software. The pseudo-temporal ordering gradually orders cells based on their transcriptional profile, for example, indicating the cellular differentiation degree. 

### 2.3. Determining DEGs

Differences in gene expression between clusters were identified using a significance analysis of the sequencing data (SAMseq) [46] from the samr package [47]. SAMseq is a non-parametric statistical function dependent on the Wilcoxon rank statistic that equalizes the sizes of the library by a resampling method accounting for the various sequencing depths. The analysis is implemented over the pure raw dataset that has the unnormalized expression read counts after excluding the ERCCs. The SAMseq function indirectly calls a subfunction called rankcol, which was written in Fortran. This function is used to attribute column-wise ranks to a matrix. Due to some limitations in handling large datasets, the function was rewritten in R and incorporated into DIscBIO. Furthermore, binomial counting statistics are used to identify the differentially expressed markers for each cluster. This is done by highlighting the DEGs in the target cluster comparing to all the remaining clusters using binomial differential expression. Differentially expressed markers help characterize the molecular signatures. For instance, when analyzing cancer cells, it might help to characterize the tumorigenic capabilities of each cell cluster, leading to the development of better therapeutics [48]. The binomial differential expression was computed using RaceID through DESeq2 [49]. Volcano plots are used to visualize the results of the differential expression analyses. 

### 2.4. Identifying Biomarkers

The biomarker discovery approach starts with a list of DEGs. Next, protein–protein interaction (PPI) networks and gene enrichment analyses are implemented to further explore the hub genes with the most interactions in gene modules [50,51]. The outcome is used to test the connectivity degree and the betweenness centrality of the interaction network, which reflects the communication flow in the networks. Additionally, decision tree analysis is implemented either between two clusters of interest or between one cluster verses the remaining cells. Putting all the results together can highlight panels of biomarkers for a particular cluster with certain characteristics. DIscBIO uses decision trees and hub detection through networking analysis and gene enrichment analysis to discover biomarkers. Decision trees are a very efficient classification technique in biomarker discovery. In the current version of DIscBIO, two different decision trees, J48 and RPART, can be implemented to predict the sub-population of a target cell based on transcriptomic data. The J48 tree is implemented through the RWeka package whereas the RPART tree is computed using the rpart package. The performance of the generated trees can be evaluated for error estimation by a ten-fold cross validation assessment. To identify the protein–protein interactions, we use STRING through its application programming interface. Moreover, to gain mechanistic insight into the DEGs, gene and pathway enrichment analysis can be performed in DIscBIO through the enrichR package. Further information on how to use DIscBIO and the possible options and documentation of the algorithms are included within our Jupyter notebooks.

## 3. Pipeline Extension

Currently, DIscBIO provides limited options for normalization, clustering, differential analysis and cell type identification. However, its open, versatile and dynamic structure enables users with programming skills to fully edit the pipeline and include other methods. Through the Jupyter notebook, we showcase how additional methods, such as Leiden clustering, can be added to the pipeline. The Leiden clustering algorithm has the ability of identifying high-quality partitions [52,53]. It does not require a cluster number a priori. The notebook is available at https://nbviewer.jupyter.org/github/ocbe-uio/DIscBIO/blob/dev/notebook/DIscBIO-CTCs-Binder-Leiden-Clustering.ipynb. Importantly, the features of DIscBIO can be expanded in future work to include other methods, such as SCDE and scTyper. SCDE is an alternative tool for differential analysis [54]. It was developed by the Kharchenko lab specifically for single-cell RNA-seq data. SCDE uses Bayesian probability and can handle the technical artifacts of single-cell RNA sequencing. RaceID, which is the fundamental tool in DIscBIO, is well known for identifying rare cell types with limited efficiency in the absence of rare cell populations [55]. On the other hand, scTyper is an R package for reproducible and comprehensive cell typing, equipped with 213 cell marker sets collected from the literature [56]. Incorporating such tools in DIscBIO can enable users to compare the differential analysis outcome from tools designed for bulk RNAseq (SAMseq) and scRNAseq (SCDE) in addition to identifying cell types in heterogeneous cell populations, with and without rare cell populations.

## 4. Case Studies

To showcase DIscBIO, we analyzed two scRNA-seq datasets: one consisting of circulating tumor cells (CTCs) from patients with breast cancer, and a second dataset from a myxoid liposarcoma cell line. Additionally, we illustrate how to use the conquer repository by loading and analyzing one of the available mice scRNA-seq datasets. For each case study, we provide a Jupyter notebook that includes the complete analysis and can be used as a guide to examine other datasets.

### 4.1. CTC Case Study

Here we analyze a dataset consisting of single (not clustered) CTCs collected from patients with breast cancer and obtained from several studies [57,58,59,60,61,62,63,64,65] . Data are available in the GEO database with accession numbers GSE51827, GSE55807, GSE67939, GSE75367, GSE109761, GSE111065 and GSE86978. The raw data includes RNA-seq from single CTCs and CTC clusters that were obtained from the blood of breast cancer patients. Here, the CTC clusters were excluded, and a dataset of 1462 single CTCs was used in the analysis. The CTC dataset was formatted in a data frame with “.csv” and “.rda” extensions. Columns refer to samples and rows refer to genes. The dataset can be inserted into DIscBIO using the “read.table” or “read.csv” functions. The CTC notebook is available at https://nbviewer.jupyter.org/github/ocbe-uio/DIscBIO/blob/dev/notebook/DIscBIO-CTCs-Notebook.ipynb. CTCs are a rare subset of cells found in the blood of cancer patients and arise as a consequence of tumors shedding cancer cells into blood vessels. They preserve primary tumor heterogeneity and imitate tumor characteristics [66]. CTCs may serve as metastatic seeds and succeed initiating secondary tumors. An increasing number of studies on CTCs focus on efforts to uncover and characterize molecular features of CTCs that predict their metastasis-generating potential. A prominent example is epithelial–mesenchymal transition [67]. Due to the widely accepted role of CTCs in metastasis, they can be used to showcase and explore potential involvement in molecular or cellular alterations in cancer progression. In our current case study, we aimed at (1) characterizing the sub-populations of CTCs, and (2) linking alterations of the Golgi apparatus with cancer progression, to showcase how our analysis pipeline can be used to study this link based on current knowledge.

#### 4.1.1. Characterization of CTC Subpopulations

To characterize sub-populations of CTCs, the dataset was filtered based on the median expression of all the genes in the dataset. Genes with less expression than the overall median expression in at least 10% of the cells were excluded. Cells with less than 1000 read counts were removed. In total, 215 genes and 1448 cells were used for further analysis. After filtering and computing gap statistics through RaceID, four clusters were obtained based on k-means clustering (Figure 2a). The clustering outcome can serve as a guide for assessing the robustness of the clusters, computing the pseudo-time ordering and detecting differentially expressed genes (DEGs). Clusters 2 and 4 were stable and consistent. Clusters 1 and 3, instead, had a low degree of stability and consistency. Pseudo-time ordering showed a clear separation between Clusters 1 and 4 (Figure 2b). Likewise, the heatmap portrayal of cell-to-cell distances, where the cluster centers were ordered by hierarchic clustering (Figure 2c), also exhibited a clear separation between Clusters 1 and 4.

Guha et al. (2018) have reported that metastasizing cells from a highly aggressive tumor exhibit a higher frequency of mitochondrial defects. To investigate the mitochondrial defects across the clusters, we plotted the expressions of two genes reported by Guha et al.: epithelial splicing regulatory protein (*ESRP1*) and mitochondrial transcription factor A (*TFAM*). Both genes were downregulated in Cluster 4 compared to their expression in Cluster 1 (Figure 2d,e). The DEGs were extracted by a two-class unpaired response test using SAMseq, with a significantly false discovery rate (FDR) less than 0.05. The list of detected DEGs is the starting point of the biomarker discovery. The gene enrichment analysis and the networking analysis performed over the list of the upregulated DEGs in Cluster 4 showed that many of these genes are involved in the processes of relevance to cancer progression, such as exocytosis regulation, Rap1 signaling, and regulation of cell migration.

The networking analysis suggested five genes, namely, *PF4*, *PPBP*, *ITGA2B*, *FERMT3* and *SELP*, as the central hub nodes (Figure 3a). The platelet factor 4 (*PF4*) is an endocrine factor with overexpression, associated with low survival of patients with lung cancer [68]. The Pro-Platelet Basic Protein (*PPBP*) is a member of the CXC subfamily of chemokines. *PPBP* was reported as an enhancer of the invasive ability of breast cancer cells [69]. Integrin alpha 2b (*ITGA2B*) is a member of the integrin family that regulates a diverse set of cellular processes and is crucial to the initiation, progression and metastasis of solid tumors, including breast cancer. The over-expression of *ITGA2B* promotes proliferation and invasion in breast cancer [70]. The *FERMT3* gene encodes a protein involved in integrin activation. *FERMT3* is known to enhance breast cancer progression and metastasis [71]. Selectin-P (*SELP*) was found to be important in organ-specific metastatic dissemination of breast cancer [72]. 

Moreover, binomial differential expression was performed to detect differentially expressed markers for each cluster. The metastasis associated lung adenocarcinoma transcript-1 (*MALAT1*) was detected to be a marker for Cluster 4. *MALAT1* has been described as a long non-coding gene with contradictory functionality. Previous studies demonstrated that *MALAT1* promotes cell proliferation, migration, tumor growth, metastasis and chemoresistance. On the other hand, a recent study by Kim and colleagues showed that *MALAT1* levels inversely correlate with breast cancer progression and metastatic ability in transgenic, xenograft and syngeneic models [73]. All the CTCs in the selected dataset showed upregulation of *MALAT1*, especially in Cluster 4 (Figure 2f). Further investigation is needed to confirm the role of *MALAT1* in circulating breast cancer cells. The overall impact of our findings reported here is that cells in Cluster 4 seem to have highly aggressive characteristics of invasion and metastasis comparing to less aggressiveness in Cluster 1. Decision tree analysis was performed using the total list of the binomial DEGs to identify the potential genes to predict the cluster identity of a target cell, and whether it belongs to Clusters 1 or 4. The generated RPART decision tree (Figure 3b) included four decision nodes: *MALAT1*, *SYNE2*, *SLC25A39* and *CFLAR*. The performance metric of the RPART decision tree is remarkably high, with an accuracy of 99%, specificity of 100% and sensitivity of 99%. The time for running the DIscBIO–CTCs–Binder notebooks (Parts 1–3) is 15 min, whereas the approximate time for running through the Jupyter notebook, the analysis described above, using an Intel Core i5-8300H laptop, is about 3 h. The time difference is due to structural deviation. Some objects, such as SC and DATAforDT, were precomputed and inserted into the Binder notebooks. 

#### 4.1.2. Linking Alterations of the Golgi Apparatus with Cancer Progression

The Golgi apparatus is an evolutionary-conserved cellular organelle that plays a key role in protein sorting and post-translational modification, as well as being a hub for signaling molecules and to thereby contribute to the outcome of signaling cascades [74]. In addition, the Golgi has a role in regulating cell migration and cell polarity [75,76]. Aberrant functional organization of the Golgi is often linked to cancer. A recurrent theme in the scientific community is the hypothesis that alterations of the Golgi structure are associated with cancer progression [77]. A wide range of genes induce structural alteration of the Golgi apparatus [78,79,80]. These alterations fall into two main categories: compaction and fragmentation of the Golgi. The available collection of genes that affect the Golgi structure might form a framework for analyzing the connection between Golgi alterations and cancer. The main challenge with genes that are linked to Golgi alterations is that they often form loosely connected networks, making it complicated to extract meaningful information with respect to enrichment of signaling pathways or metabolic processes. 

Here, the CTC dataset was used to link Golgi alterations with cancer progression based on a list of 164 genes that are known to cause Golgi fragmentation (Appendix A). These 164 genes were considered to filter the CTC dataset. From 164, only 97 genes were expressed. Silencing each of these genes is known to cause Golgi fragmentation [79,80,81,82,83]. Using the k-means clustering approach based on these 97 genes, cells were classified in four stable clusters (Figure 4a). Clusters 3 and 4 are highly consistent. Clusters 1 and 2, instead, have a low degree of consistency. 

Both pseudo-time ordering (Figure 4b) and the heatmap portrayal of cell-to-cell distances, where cluster centers were ordered by hierarchic clustering (Figure 4c), show a clear separation between Clusters 3 and 4. Based on the binomial differential expression, cells in Cluster 3 express significantly less *DGKD* and *PTCRA* but significantly more *KRT18* (Figure 4d,f). Interestingly, silencing the diacylglycerol kinase, delta (*DGKD*) has been validated to cause a condensed Golgi phenotype (Chia et al., 2012). The Pre T Cell Antigen Receptor Alpha (*PTCRA*) has recently been nominated as a biomarker in breast cancer [84]. Keratin 18 (*KRT18*) was reported as significantly less expressed in all basal-like cell lines, which are highly aggressive compared with luminal cell lines [85]. Low *KRT18* expression in breast cancer has been suggested to correlate with poor prognosis and to be a marker for epithelial–mesenchymal transition (EMT) [86,87]. Elucidating the biological significance of these findings requires further experiments. However, we might speculate that the high *KRT18* in Cluster 3 could indicate that cells in this cluster are at earlier stages of EMT and that these cells have a compact Golgi due to low *DGKD*.

The expression profiling of these 97 genes that fragment the Golgi showed two main Golgi fragmentation subsets within the Golgi genes. The first GF-subset of genes (56 genes) demonstrated upregulation in Cluster 3. The second GF-subset (15 genes) demonstrated upregulation in Cluster 4. The remaining genes (26 genes) showed either a stable low expression or unclear expression pattern across all clusters (Figure 5a). This could suggest that cells in Cluster 4 exhibit a high likelihood of Golgi fragmentation whereas cells in Cluster 3 exhibit a high likelihood of Golgi condensation. DEGs between clusters 3 and 4 were extracted by a two-class unpaired response test using SAMseq (FDR < 0.05 and fold change > 1). Five genes (*RPS27A*, *TUBA1B*, *RPL4*, *RPL34* and *RPS12*) have been classified as significantly upregulated in Cluster 3, whereas 748 genes were significantly upregulated in Cluster 4 (Appendix A). During the dissemination of breast cancer cells, they exhibit a loss of epithelial characteristics, which routinely is accompanied by upregulation of mesenchymal genes. This process is known as epithelial–mesenchymal transition (EMT). Several studies highlighted a strong association between the expression of EMT genes in circulating breast cancer cells and cancer progression, invasion and metastasis [88,89]. To investigate this association, we profiled the expression of 20 genes (Figure 5b). These genes were nominated as EMT markers by Zhao and colleagues [90]. Seven EMT genes (*AKT1*, *EGFR*, *EPAS1*, *ERBB2*, *HIF1A*, *SMAD3* and *MET*) were relatively upregulated in Cluster 3 comparing to Cluster 4. 

However, four genes (*CTNNB1*, *ILK*, *TGFB1* and *ZEB2*) were classified as significantly upregulated in Cluster 4; these genes exhibit similar expression patterns to the genes in the second GF subset. *TGFB1* is a major inducer of EMT and is commonly used to induce EMT for research purposes [91]. *TGFB1* is highly expressed in all clusters except Cluster 3. Interestingly, we found a fairly strong negative correlation (r = −0.67, *p* < 0.001) between *TGFB1* and *KRT18*, which is in line with previous findings [86]. Our data suggest that Golgi fragmentation is correlated with markers of EMT. On the other hand, others reported that EMT derives Golgi compaction through a strong correlation between *MMD* and *Zeb1* [92]. However, we found a weak negative correlation between the expression of *MMD* and *Zeb1* (r = −0.15, *p* < 0.001). This discrepancy highlights the complexity of the link between Golgi structural alterations and cancer progression and indicates the need for more experimental efforts to clarify the role of the Golgi in cancer. The time for running the DIscBIO–CTCs–Binder notebooks (Parts 4 and 5) was 12 min, whereas the approximate time for running through the DIscBIO package, the analysis described above, using an Intel Core i5-8300H laptop, is about 2 h. The time difference is due to structural deviation. Some objects, such as fg and FgcdiffBinomial, were already precomputed and inserted into the Binder notebooks.

### 4.2. MLS Case Study

Here, we analyze scRNA-seq data from a myxoid liposarcoma (MLS) cell line. The MLS data, including ERCC spike-ins and 94 single MLS cells with 59838 genes, are available in the ArrayExpress database at EMBL-EBI with accession number E-MTAB-6142 [93]. The MLS notebook is available at https://nbviewer.jupyter.org/github/ocbe-uio/DIscBIO/blob/dev/notebook/DIscBIO-MLS-Binder.ipynb. Myxoid liposarcoma is a rare type of tumor driven by specific fusion oncogenes, normally FUS-DDIT3 [94,95], with few other genetic changes [96,97]. The 94 single cells were collected based on their cell cycle phase (G1, S or G2/M), and assessed in the collection step by analyzing their DNA content using Fluorescence Activated Cell Sorter [93]. The original study aimed at investigating the gene expression dynamics during the course of the cell cycle in MLS cells. Moreover, it aimed at resolving the dynamics within phases using pseudo-time ordering as well as gene clustering of the DEGs between phases [93]. Here, we take another direction aiming at defining the molecular signature of sub-populations of MLS cells in association with their cell cycle phases. After downloading, unzipping the “E-MTAB-6142.processed.1.zip” and saving it in the working directory, the read counts matrix can be loaded to DIscBIO using the “read.table” or “read.csv” functions. During the preprocessing, genes were selected by accounting for technical noise based on the variation and expression of the ERCC spike-ins, which were added to each sample before library preparation (Figure 6a). A total of 5684 genes (black dots) have a variation above the noise level (red curve). The resulted genes from the technical noise filtering were used for the downstream analysis. After filtering the dataset, three stable MLS clusters (Jaccard similarity > 0.6) were obtained based on model-based clustering and visualized in a PCA plot (Figure 6b). Clusters can be visualized by *t*SNE maps as well. However, considering the distinctive separation, PCA plots showed intact projection of the contents of the clusters generated by model-based clustering. One possible explanation could be that the linear probabilistic processing of the PCA matches with the probability model-based clustering, which generates a model for every cluster and computes the best fit of the data to the generated model [98,99]. The MLS clusters match with the cell cycle phases in the initial study [93]. Cluster 1 is characterized by low consistency (Silhouette width = −0.1), whereas Clusters 2 and 3 have reasonable consistency (Silhouette width > 0). Pseudo-temporal ordering showed a gradual transition between the cells’ transcription profiles across the clusters (Figure 6c). Both the clustering and pseudo-time overlap with the cell cycle phases (Figure 6d). Cells in Cluster 2 are mainly in the G2 phase, whereas the majority of the cells in Cluster 3 are in the S phase. Cluster 1 shows heterogeneous cells with different cell cycle phases. No outlier cells were detected in any of the clusters (Figure 6e). To investigate the similarities between the single cells, Euclidean distances of Pearson transcriptome correlation matrix were computed. Based on these similarities, a heatmap portrayal of the cell-to-cell distances was plotted using Euclidean as the distance measure and single linkage as the clustering method.

To further define the molecular signature of the clusters, SAMseq and binomial differential expression analyses were performed (FDR < 0.05). Significant DEGs were highlighted in the volcano plots (Figure 7a). Moreover, The DEGs of each cluster underwent PPI network construction. The network of the downregulated genes in Cluster 2 (Figure 7b) highlighted thymidylate synthetase (*TYMS*) as a hub gene in addition to a set of highly connected genes involved in DNA synthesis and replication. *TYMS* has been reported to be associated with the well-differentiated subtype of liposarcoma [100]. *TYMS* is upregulated in Cluster 3 (Figure 7c). To identify potential genes to predict the cluster identity of a target cell, and whether it belongs to Cluster 2 or 3, decision tree analysis was implemented. The generated RPART decision tree (Figure 7d) included two decision nodes: Aurora Kinase A (*AURKA*) and Cell Division Cycle 27 (*CDC27*). *AURKA* is involved in self-renewal of breast cancer stem cells [101]. *CDC27* is known to induce metastasis, invasion and sphere-formation in colorectal and gastric tumors [102,103]. Cluster 2 cells, which were ordered in the foreground pseudo-temporal ordering, highly express genes involved in regulating proliferation, EMT, stemness and chemoresistance acquisition in several cancers. Stemness is known to be frequently associated with quiescence [104]. Several studies reported that the decision for entering quiescence is facilitated during the maternal G2 phase [105,106]. G2 arrest in stem cells is associated with robust regeneration capacity and it has been reported to be reversible and released when regeneration must be achieved [107]. The networking analysis outcome of the DEGs in Cluster 2 (Appendix A) nominated a panel of 18 hub genes, including *PLK1*, *CDC20*, *KIF2C*, *AURKA, BUB1*, *AURKB* and *PTTG1* as potential biomarkers for Cluster 2 cells. The overall impact of our findings reported here is that cells in Cluster 2, which are mainly in the G2 phase, seem to exhibit aggressive and stem-like properties and further computational and experimental investigations should validate these biomarkers candidates in myxoid liposarcoma. The approximate time for running the DIscBIO–MLS–Binder notebook is 15 min. The same time is needed to run the DIscBIO package for the analysis described above using a laptop (Intel Core i5-8300H).

## 5. A Comparative Analysis of DIscBIO against Similar scRNAseq Pipelines

Many of the tools used in DIscBIO were benchmarked before [11,55,100,101]. We compared DIscBIO with other existing, fully-integrated web-based scRNAseq pipelines, including Granatum [16], ASAP [15] and SC1 [17] (Appendix A). Unlike the other systems, DIscBIO can be either used online through Jupyter notebooks or downloaded and run locally in command-line mode. All tested tools accept gene expression matrices as input in .csv format. ASAP and SC1 support 10x files as well. Granatum is the only pipeline providing basic batch-effect removal capability using either ComBat or median alignment. Gene filtering is a very critical step of each pipeline due to its impact on the accuracy and efficiency of the downstream analysis. Most workflows filter genes based on the overall expression levels. Moreover, DIscBIO also enables users to filter genes based on technical noise using ERCC spike-ins or depending on a desired panel of genes involved in a particular pathway/biological process, or all the three techniques combined. All tools offer several clustering methods, with k-means being a common method. However, Granatum, ASAP and SC1 do not support evaluating the robustness of the detected clusters. On the contrary, DIscBIO enables robustness assessment of the stability and consistency of the detected clusters using Jaccard’s similarity statistics and silhouette coefficients. All pipelines allow users to select multiple options for downstream differential expression and pathway enrichment analyses. Notably, DIscBIO also includes a networking analysis to detect the hub genes linked to the results of the pathway enrichment analysis. Moreover, unlike other systems, DIscBIO implements decision trees to detect biomarkers.

We performed a side-by-side evaluation of the performance of DiscBIO and Granatum using the CTC dataset (Appendix A). With Granatum, we normalized the read counts based on the quantile. Genes were filtered based on the Log Mean Expression Threshold and Dispersion Fit Threshold. With DIscBIO the read counts were normalized using the median of the ratios. Genes were filtered based on the minimum expression in a certain number of cells. Cells were clustered in four clusters using k-means (Euclidean). About 43% of the cells were clustered similarly to Granatum and DIscBIO. We evaluated the stability and consistency of the clusters generated using Granatum by downloading the cluster IDs and a matrix of filtered expressions and inserting them into DIscBIO. The robustness assessment of the generated clusters shows a similar stability between the Granatum and DIscBIO clusters, but with a much higher consistency in the DIscBIO clusters. We also performed differential expression analysis using the two pipelines and observed similar panels of DEGs. The overall comparison results are available as a notebook at https://nbviewer.jupyter.org/github/ocbe-uio/DIscBIO/blob/dev/notebook/DIscBIO_VS_Granatum_%20Notebook.ipynb.

## 6. Conclusions

To reduce the complexity of single-cell transcriptomics analyses using a combination of specific computational tools for scRNA-seq analysis, we developed DIscBIO as a step-wise approach. It is available as an R package, Jupyter notebook and Binder cloud version. This makes it a convenient tool for a variety of researchers to facilitate the exploration of their scRNA-seq data. The open, versatile and dynamic structure of DIscBIO enables users with programming skills to fully edit the pipeline and extend it to include other methods. Furthermore, and due to its sequential narrative design and explanatory structure, DIscBIO can be used for teaching and training purposes. Using DIscBIO we were able to identify the CTCs with highly aggressive characteristics of invasion and metastasis in breast cancer and also gain insights about the link between the alterations of the Golgi apparatus and cancer progression. Furthermore, DIscBIO enabled us to identify a small subset of cells with possible aggressive and stem-like properties in myxoid liposarcoma.

## Figures and Tables

**Figure 1 ijms-22-01399-f001:**
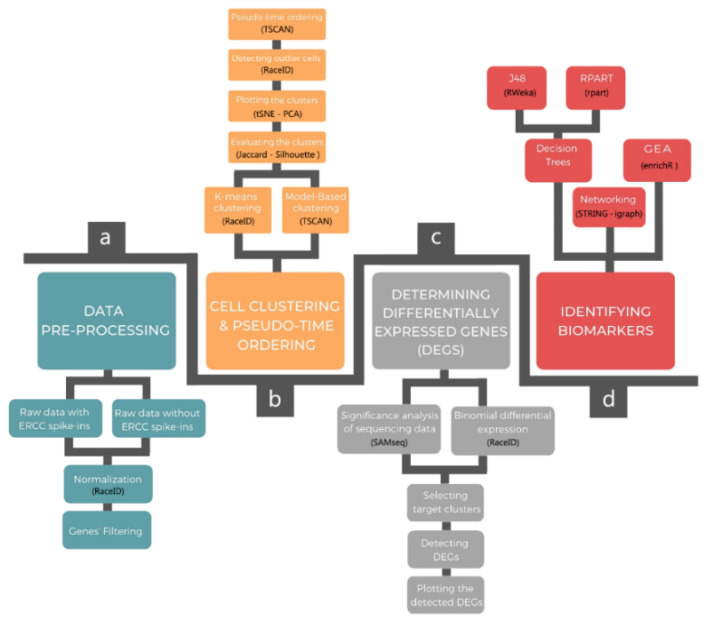
Overview of the DIscBIO pipeline showing its four successive sections. (**a**) Pre-processing the raw read counts with and without ERCC spike-ins. (**b**) Cell clustering and pseudo-time ordering. During this step clusters can be evaluated and visualized in PCA plots and *t*SNE maps with a possibility to detect outlier cells. (**c**) Identifying differentially expressed genes by SAMseq and binomial counting statistics. (**d**) Identifying biomarkers through decision trees, networking and gene enrichment analysis.

**Figure 2 ijms-22-01399-f002:**
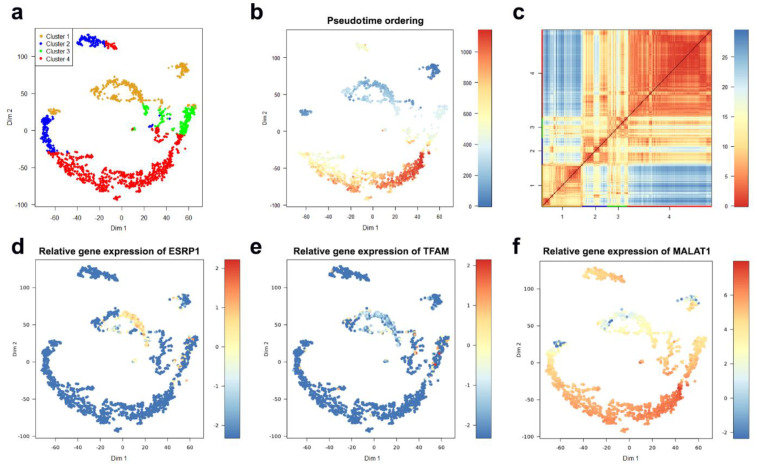
Identification of the circulating tumor cell (CTC) sub-populations, clustered based on the median expression of all genes in at least 10% of the cells. (**a**) *t*SNE map visualizing clusters of circulating breast cancer cells identified with k-means clustering. (**b**) The plot from (**a**) with cells colored based on their pseudo-time ordering. (**c**) A heatmap portrayal of cell-to-cell distances; cluster centers were ordered by hierarchic clustering. (**d**) The plot from (**a**) with cells colored based on their *ESRP1* expression. (**e**) The plot from (**a**) with cells colored based on their *TFAM* expression. (**f**) The plot from (**a**) with cells colored based on their *MALAT1* expression.

**Figure 3 ijms-22-01399-f003:**
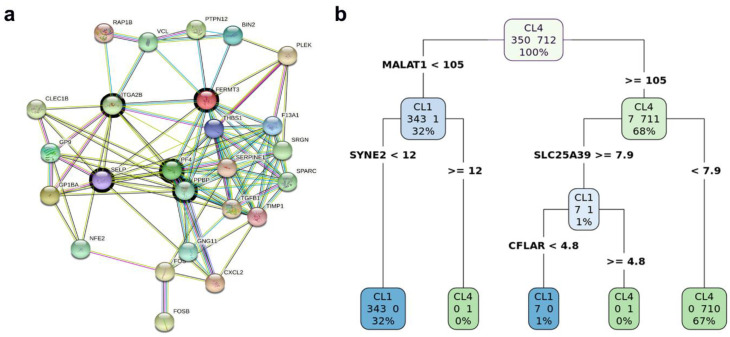
Biomarker discovery in Cluster 4. (**a**) A sub-network of upregulated DEGs in Cluster 4. (**b**) A schematic figure explaining the RPART decision tree to predict the cluster identity of a target cell and whether it belongs to Cluster 1 or 4.

**Figure 4 ijms-22-01399-f004:**
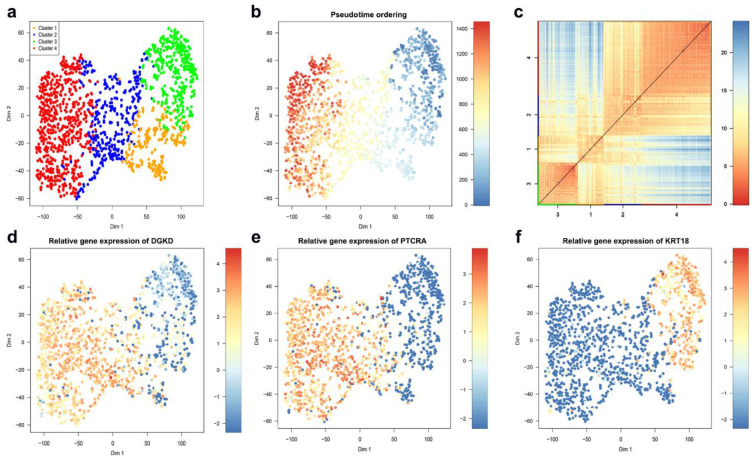
Identification of the CTC sub-populations, clustered based on the Golgi fragmentation gene list. (**a**) *t*SNE map visualizing clusters of circulating breast cancer cells identified with k-means clustering. (**b**) The plot from (**a**) with cells colored based on their pseudo-time ordering. (**c**) A heatmap portrayal of cell-to-cell distances; cluster centers were ordered by hierarchic clustering. (**d**) The plot from (**a**) with cells colored based on their *DGKD* expression. (**e**) The plot from (**a**) with cells colored based on their *PTCRA* expression. (**f**) The plot from (**a**) with cells colored based on their *KRT18* expression.

**Figure 5 ijms-22-01399-f005:**
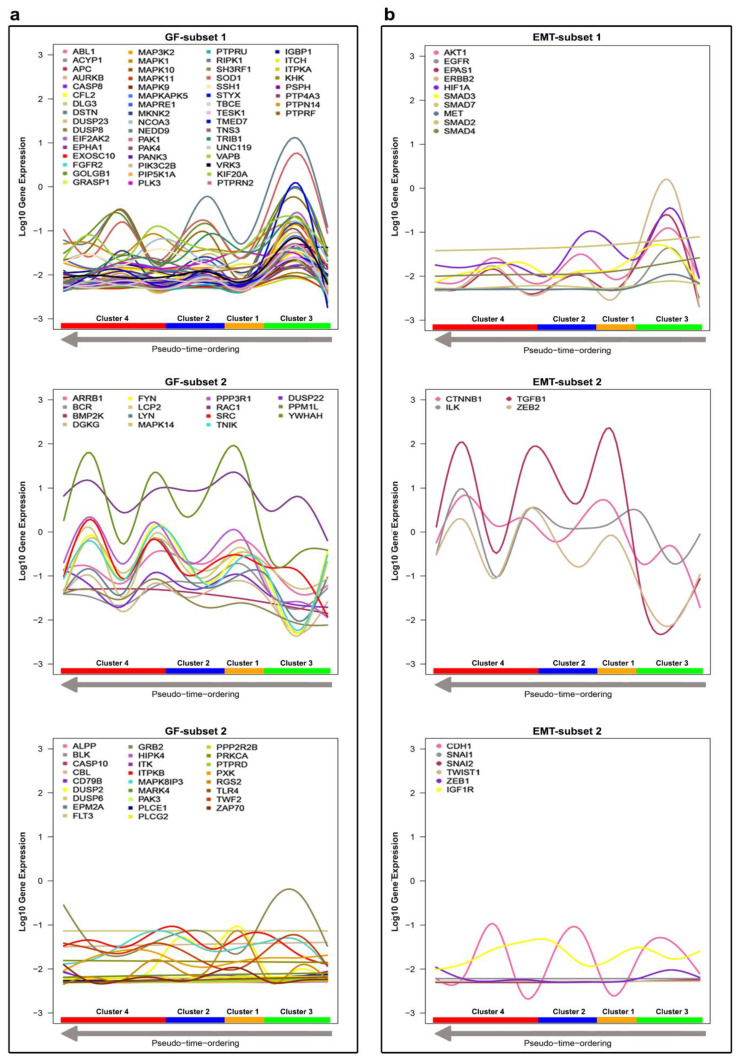
Discovery and exploration of the patterns in gene expression across the clusters, ordered from right to left based on pseudo-time ordering. The log10 gene expression values are shown on the vertical axis. Generalized additive mode smoothing is used to smooth the expression values per gene. (**a**) Gene expression profiling of the Golgi fragmentation gene list (97 genes). (**b**) Gene expression profiling of the EM gene list (20 genes).

**Figure 6 ijms-22-01399-f006:**
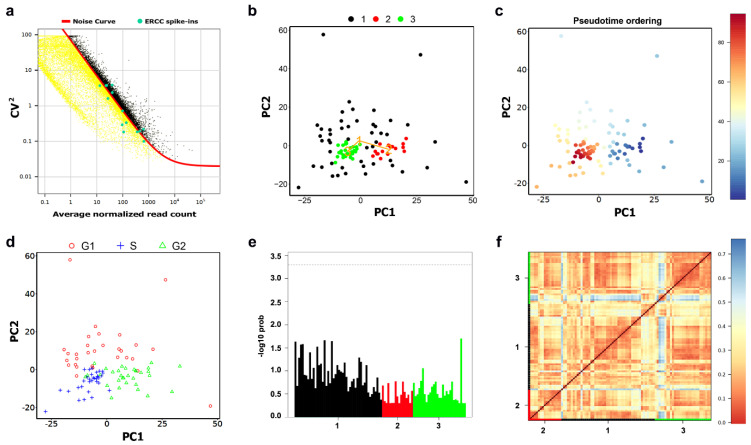
Identification of the MLS sub-populations. (**a**) Gene filtration by accounting for technical noise based on the variation and expression of the ERCC spike-ins. (**b**) PCA plot visualizing clusters identified using model-based clustering. (**c**) The plot from (**a**) with cells colored based on their pseudo-time ordering. (**d**) The plot from (**a**) with cells labeled based on their cell cycle phase. (**e**) A bar-plot of the outlier probabilities of all cells across clusters. (**f**) A heatmap portrayal of cell-to-cell distances; cluster centers were ordered by hierarchic clustering.

**Figure 7 ijms-22-01399-f007:**
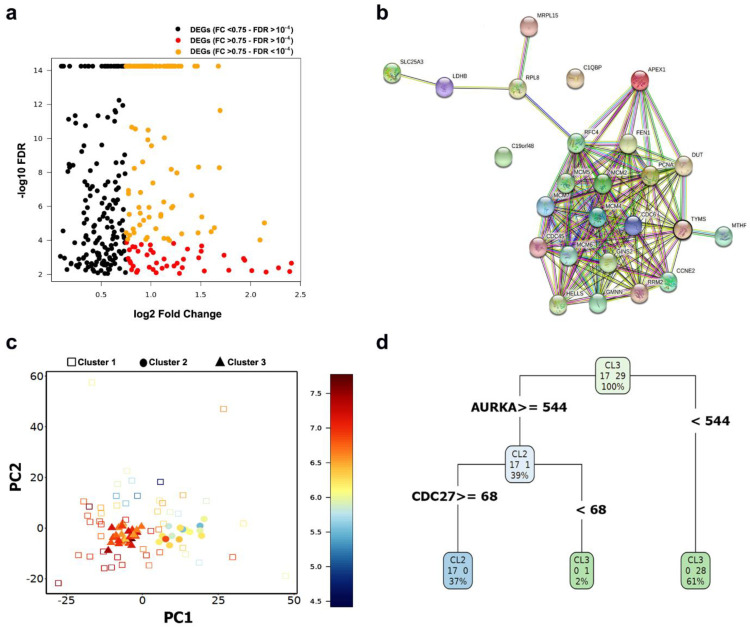
Biomarker discovery in Cluster 2. (**a**) Volcano plot showing the upregulated markers in Cluster 2. These markers were detected using binomial differential expression. (**b**) Network of downregulated DEGs in Cluster 2. (**c**) PCA plot visualizing cells labeled based on their cluster ID and color-coded based on the expression of *TYMS*. (**d**) Schematic figure explaining the RPART decision tree to predict the cluster identity of a target cell to belong to Cluster 2 or 3.

## Data Availability

DIscBIO is an open-source pipeline whose source code and Jupyter notebooks are deposited in a GitHub repository https://github.com/ocbe-uio/DIscBIO. Its CRAN-published R package can be downloaded from https://cran.r-project.org/web/packages/DIscBIO/index.html or installed directly from a regular R session. DIscBIO is also provided as a cloud version using Binder. Instructions to run our Binder notebooks can be found in the README on our GitHub repository.

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
