# Peer review of "DIscBIO: A User-Friendly Pipeline for Biomarker Discovery in Single-Cell Transcriptomics"

_ijms, 2021, doi:10.3390/ijms22031399_

Round 1

Reviewer 1 Report

Summary:

The author presents a bioinformatic pipeline DIscBIO that integrates existing packages to perform standard single-cell RNA sequencing analysis, which includes preprocessing, pseudotemporal ordering, differential gene expression, and biomarker identification to aid the computationally challenged researchers. The pipeline is available both as an R package as well as a cloud version using Binder. Two case studies were presented using the pipeline.

Major comments:

1. The pipeline aims to facilitate the computationally challenged researchers to perform the standard scRNAseq analysis. However, from the Jupyter notebooks provided, I failed to see how this is a simplification of the procedures as it requires an equally challenging understanding of basic programming and parameter optimization.

2. There are similar pipelines that provide integrated scRNAseq standard processing, such as scTyper, Granatum, and SC1. A benchmarking against those current approaches is absent from this manuscript.

3. The options for clustering is limited. It would be beneficial to include clustering options such as Louvain clustering and Leiden clustering that do not require a cluster number a priori.

4. It is unclear how the biomarkers are identified/discovered. More elaborations are needed.

5. What is the average time for running a full analysis through DIscBIO?

Minor comments:

1. Since the data from two case studies were obtained from published studies, it would be interesting to emphasize the similarities and differences between the current reanalysis and the original study.

Author Response

Response to Reviewer 1 Comments

Point 1: The pipeline aims to facilitate the computationally challenged researchers to perform the standard scRNAseq analysis. However, from the Jupyter notebooks provided, I failed to see how this is a simplification of the procedures as it requires an equally challenging understanding of basic programming and parameter optimization.

 Response 1: Thank you for highlighting this, and apologies we were not clearer in the original manuscript. The Jupyter notebooks we provide show a sequential workflow of our pipeline, where all used packages are integrated. Following the explanatory text, R code and output images and data images included in the notebooks, R users can understand the different steps of the pipeline and learn how to apply it to analyse their own data. Doing so, they do not need to worry about the non-trivial task of connecting the different software. But in order to do meaningful analyses, they still need to understand the implemented methods, their possible options or parameters and their limitations. We believe that users with some programming skills will be able to tune individual parameters while more experienced programmers can fully edit and extend the pipeline to their needs. We agree that the notebooks can be challenging for users with very limited programming skills, especially those that are not R users. We provide a cloud version using Binder that allows the execution of the pipeline without the need of downloading R, jupyter or any of the packages used by the pipeline. Although all users can use those, we see the Binder notebooks as a more didactic option that can be used for training purposes.

 We have modified the abstract (lines 26 and 30 to 36) and also added sentences in page 2, line 83-89 of the introduction to clarify it.

 Point 2: There are similar pipelines that provide integrated scRNAseq standard processing, such as scTyper, Granatum, and SC1. A benchmarking against those current approaches is absent from this manuscript.

 Response 2: Thank you for pointing out the absence of a benchmarking against similar pipelines. In fact, many of the tools used in our pipeline have already been benchmarked (Soneson & Robinson, 2018; Tian et al., 2019; Vieth et al., 2019; Peyvandipour et al., 2020). We are now reporting those benchmarking studies in our manuscript in page 13, line 455. A full benchmarking of DIscBIO against similar pipelines is outside the scope of this paper, as it requires extensive additional work. However, we agree that at least a comparative analysis is needed and for that purpose we added to the manuscript a new section entitled “A comparative analysis of DIscBIO against similar scRNAseq pipelines”, where we compare DIscBIO features to Granatum, ASAP and SC1. A summary of the comparison is provided in the supplementary table S2. Moreover, to investigate the performing of DIscBIO we have analysed the same dataset, the CTC dataset, with both DIscBIO and Granatum and compared results. A summary of the comparison is provided in the supplementary table S3. The overall impact of our comparison shows that both DIscBIO and Granatum mutually agreed on clustering about half of the cells and detected relatively similar panels of DEGs. The comparison is available as a binder notebook at: https://nbviewer.jupyter.org/github/ocbe-uio/DIscBIO/blob/dev/notebook/DIscBIO_VS_Granatum_%20Notebook.ipynb

  • Peyvandipour, A.; Shafi, A.; Saberian, N.; Draghici, S. Identification of cell types from single cell data using stable clustering. Scientific reports 2020, 10, 1-12.
  • Soneson, C., & Robinson, M. D. (2018). Bias, robustness and scalability in single-cell differential expression analysis. Nature methods15(4), 255.
  • Tian, L., Dong, X., Freytag, S., Le Cao, K. A., Su, S., JalalAbadi, A., ... & Naik, S. H. (2019). Benchmarking single cell RNA-sequencing analysis pipelines using mixture control experiments. Nature methods16(6), 479-487.
  • Vieth, B., Parekh, S., Ziegenhain, C., Enard, W., & Hellmann, I. (2019). A systematic evaluation of single cell RNA-seq analysis pipelines. Nature communications10(1), 1-11.

We modified the manuscript to address this point at pages 13-14, lines 454-487.

Point 3: The options for clustering is limited. It would be beneficial to include clustering options such as Louvain clustering and Leiden clustering that do not require a cluster number a priori.

 Response 3: Thank you for this suggestion. Although DIscBIO provides limited options for normalization, clustering, differential analysis and cell type identification, its versatile and dynamic structure enables users with programming skills to fully edit the pipeline and include other methods. We agree that including additional clustering methods that do not require a cluster number a priori can add extra value to DIscBIO. We have created a Binder notebook to illustrate how to extend DIscBIO by implementing the Leiden clustering over the CTC dataset. The notebook is available at: https://nbviewer.jupyter.org/github/ocbe-uio/DIscBIO/blob/dev/notebook/DIscBIO-CTCs-Binder-Leiden-Clustering.ipynb

Moreover, we added a new section to the manuscript entitled ”Pipeline extension”. The new clustering method, together with several other ongoing and future extensions, will be added to the pipeline when we will release a new version of the package in CRAN. We modified the manuscript to address this point at page 5, lines 192-209.

Point 4: It is unclear how the biomarkers are identified/discovered. More elaborations are needed.

 Response 4: Thank you for this suggestion. To elaborate how the biomarkers are identified, we expanded point 2.4. by outlining the steps of the biomarker discovery approach. We modified the manuscript to address this point at page 5, lines 175-181.

Point 5: What is the average time for running a full analysis through DIscBIO?

 Response 5: We agree that it is important to inform the reader with the estimated time for running DIscBIO pipeline over different datasets through the different platforms of DIscBIO, including Jupyter notebooks, Binder notebooks and R. Such information has been added to the manuscript at the following positions:

Page 7, lines 286-289

Page 11, lines 375-379

Page 12, lines 444-446

Minor comments:

Point 1: Since the data from two case studies were obtained from published studies, it would be interesting to emphasize the similarities and differences between the current reanalysis and the original study.

Response 1: We thank the referee for pointing out the absence of the similarities and differences between our analysis and the original study. In both case studies, the MLS and the CTC, we took another direction from the original study analysis. We highlighted the aim of the original study at the start of the corresponding case study. Such information has been added to the manuscript at the following positions:

Page 6, lines 232-236

Page 11, lines 388-394

 Once again, we thank you for the time you put in reviewing our paper, for the constructive criticism and we look forward to meeting your expectations.

 Reviewer 2 Report

In this research article, the authors are highlighting the complexity of embedding multiple tools to perform single-cell analysis for non-trained bioinformatician. The authors are presenting their solution to this issue: a pipeline allowing data pre-processing, cell clustering and pseudo-time ordering, determination of DGE and a visualization module based on protein interaction and decision tree. To illustrate the pipeline, the authors used two data-sets: one data-set published by an independent research and one data-set published previously by the co-authors. The authors conclude their report by highlighting that the pipeline is available from multiple platforms and can be used for training/teaching purposes.

Overall, this manuscript provides a very useful workflow as well as a very good description of the tool purpose and output. We think that the originality of the manuscript is the two detailed examples which provide to the reader a framework for analysis.

However, the rationale behind the use of the specific tools used in the workflow, as well as the review of the current state of the workflow already available, are critically missing in this manuscript. Without this information, much research work will still be needed by the researchers willing to use the pipeline.

More precisely:

  1. The description of the data pre-processing is not detailed enough. It would be important for the reader that the authors describe which tools are used for which step and based on which rational. This is particularly important here as the manuscript is aimed to users who are not necessarily skilled in programming and who might not know basic processing steps.
  2. A description of alternative tools is needed. It should contain at least a well-argued explanation of the tool chosen.
  3. Similarly, a description of a few alternative pipelines already existing is needed. We are aware that very few pipelines are as complete as DIscBIO, but the manuscript should inform the reader of them.
  4. It would be helpful to have the tools used indicated in the figure 1 (such as SAMtool in the DEGS). Sub-numbers in the figure 1 (such as 1.1 or 1.2) are not explained in the figure legend nor in the pipeline description.
  5. In the description of the example, it would be informative to mention which tools provided which output. This would help the readers to understand why it is critical to have a workflow integrating the different tools.
  6. Both examples used as illustration are based on differential gene expression in relatively homogeneous cell population. It would be interesting to have in the discussion comments on the use of DIscBIO for cell atlas analysis.
  7. For clarity, we recommend the authors to start the paragraph 6.1.2 with mentioning that the analysis is performed on the CTC data-set.
  8. In figure 2 and following, we recommend the user another choice of color than dark blue and black for group 1 and 2. Color similarity makes particularly difficult reading the figure legend.
  9. Figure 5 needs a panel in which colors and groups are resumed to avoid flipping from figure 4 and 5.
  10. We advise the authors to follow the structure of the chapter 6.1 in the chapter 6.2 to simplify the  and introduce at the start of the chapter the reference of the second data-set (old reference 74).
  11. In the discussion, it would be interesting for the reader to have examples of other tools that are used in single cell data interpretation, either on visualization such as such as pathway database, or on cell type identification database, or in other topics such as data exchange. We think that disclaiming this critical point is also fair to highlight limitation of the workflow.

Author Response

Response to Reviewer 2 Comments

Point 1: The description of the data pre-processing is not detailed enough. It would be important for the reader that the authors describe which tools are used for which step and based on which rational. This is particularly important here as the manuscript is aimed to users who are not necessarily skilled in programming and who might not know basic processing steps.

 Response 1: Thank you for this suggestion. We added extra information to the pre-processing section 2.1 and included details about the used tools and parameters. We modified the manuscript to address this point at page 4, lines 127, 129-130 and 134-137.

Point 2: A description of alternative tools is needed. It should contain at least a well-argued explanation of the tool chosen.

 Response 2: We added to the manuscript in page 5, lines 192-209 a new section entitled “Pipeline extension”, where we demonstrate some alternative tools which we would like to include in DIscBIO in the near future. Regarding the argued explanation of the tool chosen, we already motivated our choice in page 2, lines 50-54, 61-64 and 66-68.

Point 3: Similarly, a description of a few alternative pipelines already existing is needed. We are aware that very few pipelines are as complete as DIscBIO, but the manuscript should inform the reader of them.

Response 3: We mentioned in the introduction in pages 1-2 lines 46-47 some other alternative pipelines, such as Seurat, SC3,ASAP, Granatum and SC1. Moreover, we added to the manuscript in pages 13-14, lines 454-487 a new section entitled “A comparative analysis of DIscBIO against similar scRNAseq pipelines”, where we compare DIscBIO features to Granatum, ASAP and SC1.

 Point 4: It would be helpful to have the tools used indicated in the figure 1 (such as SAMtool in the DEGS). Sub-numbers in the figure 1 (such as 1.1 or 1.2) are not explained in the figure legend nor in the pipeline description.

Response 4: We agree that it is better to include the different tools used in the pipeline figure. We have also added extra information of the four pipeline sections in the figure legend and in the pipeline description. Such information has been added to the manuscript at the following positions:

Pages 3-4, lines 117-123

Page 4, lines 134-137

Point 5: In the description of the example, it would be informative to mention which tools provided which output. This would help the readers to understand why it is critical to have a workflow integrating the different tools.

Response 5: Thanks for this suggestion. We highlighted in the case studies the input and the output of the different tools. Such information has been added to the manuscript at the following positions:

Page 6, lines 244-246 and 257

Page 11, line 398

Point 6: Both examples used as illustration are based on differential gene expression in relatively homogeneous cell population. It would be interesting to have in the discussion comments on the use of DIscBIO for cell atlas analysis.

Response 6: DIscBIO is mainly dependent on RaceID, which is known for being very efficient for detecting and identifying rare cell types. However, RaceID is not very efficient in the absence of rare cell populations in heterogeneous/homogeneous cell populations. We have now highlighted this limitation in the text. We modified the manuscript to address this point at page 5, lines 203-205.

Point 7: For clarity, we recommend the authors to start the paragraph 6.1.2 with mentioning that the analysis is performed on the CTC data-set.

Response 7: Thanks for this suggestion. We adjusted the structure of this section to mention directly after introducing “Glogi alterations” that the analysis is performed on the CTC dataset. We modified the manuscript to address this point at page 8, lines 316-318.

Point 8: In figure 2 and following, we recommend the user another choice of color than dark blue and black for group 1 and 2. Color similarity makes particularly difficult reading the figure legend.

Response 8: We adjusted all the figures where dark blue and black are presented together by replacing one of them with another color. Such color adjustment has been added to the manuscript at the following positions:

Page 7, figures 2a and 2c

Page 9, figures 4a and 4c

Page 10, figures 5a and 5b

Page 12, figures 6a, 6b, 6e and 6f

Page 13, figure 7a

Point 9: Figure 5 needs a panel in which colors and groups are resumed to avoid flipping from figure 4 and 5.

Response 9: Thank you for this suggestion. We added the cluster number above each cluster, in the clusters’ panel. We modified the manuscript to address this point at page 10.

Point 10: We advise the authors to follow the structure of the chapter 6.1 in the chapter 6.2 to simplify the and introduce at the start of the chapter the reference of the second data-set (old reference 74).

Response 10: Thanks for the advice. We adjusted the structure of the MLS case study so it starts with an introduction of the MLS dataset and its reference. We modified the manuscript to address this point at page 11, lines 381-383.

Point 11: In the discussion, it would be interesting for the reader to have examples of other tools that are used in single cell data interpretation, either on visualization such as such as pathway database, or on cell type identification database, or in other topics such as data exchange. We think that disclaiming this critical point is also fair to highlight limitation of the workflow.

Response 11: We are aware of the limitations of DIscBIO to identify cell types. In future work we are planning to include additional tools, such as scTyper to identify cell types in heterogeneous cell populations with and without rare cell populations. As in the revised manuscript we have added a new section on pipeline extension, we have included comments on future extensions on that section. We modified the manuscript to address this point at page 5, lines 193-195 and 205-209.

Once again, we thank you for the time you put in reviewing our paper, for the constructive criticism and we look forward to meeting your expectations.

Reviewer 3 Report

The manuscript brings the presentation of a new pipeline for use in the analysis of transcripts, which is very interesting for researchers who are users of bioinformatics tools, but have little skills in using command lines or more in-depth strategies. I believe that, if the authors add some details of how to insert the datasets into the programms still in the body of the text, it will be even more interesting for the reader.

Author Response

Response to Reviewer 3 Comments

The manuscript brings the presentation of a new pipeline for use in the analysis of transcripts, which is very interesting for researchers who are users of bioinformatics tools, but have little skills in using command lines or more in-depth strategies. I believe that, if the authors add some details of how to insert the datasets into the programms still in the body of the text, it will be even more interesting for the reader.

Response: Thank you very much for your positive evaluation of our work. We fully agree that adding details about the format of the dataset and how it is inserted into DIscBIO will be beneficial for the reader. We modified the manuscript to address this point at page 6, lines 221-223 and page 11, lines 392-394.

 Once again, we thank you for the time you put in reviewing our paper, for the constructive criticism and we look forward to meeting your expectations.

 Round 2

Reviewer 2 Report

All issues originally addressed are satisfactory answered.

New notebooks generated are very interesting. They are really helpful to use the article in a didactic context.